# DIFFER: Decomposing Individual Reward for Fair Experience Replay in Multi-Agent Reinforcement Learning

**Xunhan Hu[1], Jian Zhao[1], Wengang Zhou[1,2][†], Ruili Feng[3,1], Houqiang Li[1,2][†]**

[1] University of Science and Technology of China, [2] Institute of Artificial Intelligence,
Hefei Comprehensive National Science Center, [3] Alibaba Group
`{cathyhxh,zj140}@mail.ustc.edu.cn, zhwg@ustc.edu.cn,`
`ruilifengustc@gmail.com, lihq@ustc.edu.cn`

## Abstract

Cooperative multi-agent reinforcement learning (MARL) is a challenging task, as agents must learn complex and diverse individual strategies from a shared team reward. However, existing methods struggle to distinguish and exploit important individual experiences, as they lack an effective way to decompose the team reward into individual rewards. To address this challenge, we propose DIFFER, a powerful theoretical framework for decomposing individual rewards to enable fair experience replay in MARL. By enforcing the invariance of network gradients, we establish a partial differential equation whose solution yields the underlying individual reward function. The individual TD-error can then be computed from the solved closed-form individual rewards, indicating the importance of each piece of experience in the learning task and guiding the training process. Our method elegantly achieves an equivalence to the original learning framework when individual experiences are homogeneous, while also adapting to achieve more muscular efficiency and fairness when diversity is observed. Our extensive experiments on popular benchmarks validate the effectiveness of our theory and method, demonstrating significant improvements in learning efficiency and fairness. The code is available in https://github.com/cathyhxh/DIFFER.

## 1 Introduction

In widely adopted cooperative multi-agent systems [1–4], a team of agents typically operates under the constraint of individual local observations and limited communication capabilities. In each time step, all agents collectively interact with the environment by taking team actions, leading to a transition to the subsequent global state and the provision of a team reward. Consequently, the agents must acquire the ability to effectively coordinate their actions and maximize the cumulative team return. Over the past few years, cooperative multi-agent reinforcement learning (MARL) has demonstrated remarkable achievements in addressing such challenges and has exhibited promising potential across diverse domains, including multi-player games[5–9], autonomous cars [10, 11], traffic light control [12, 13], economy pricing [14] and robot networks [15, 16].

In recent years, value factorization methods [17–20] have emerged as leading approaches for addressing cooperative multi-agent tasks [21–23]. These methods are rooted in the centralized training and decentralized execution (CTDE) paradigm [24–26]. Specifically, each agent's network models the

---

[†]Corresponding authors: Wengang Zhou and Houqiang Li

37th Conference on Neural Information Processing Systems (NeurIPS 2023).

individual action-value using the agent's local action-observation history [27] and the last action. These individual action-values are then integrated into a global state-action value through a mixing network. The parameters of both the agent networks and the mixing network are optimized by minimizing the team TD-error. To enhance data utilization, value factorization methods commonly employ experience replay. Here, **experience** represents the atomic unit of interaction in RL, $i.e.$, a tuple of (observation, action, next observation, reward) [28]. In traditional MARL experience replay algorithms, the replay buffer stores team experiences and allows algorithms to reuse them for updating the current network, which encompasses both the agent networks and the mixing network.

However, the utilization of team experience overlooks the inherent differences among individual experiences, thereby failing to leverage individual experiences worthy of learning. This issue becomes especially pronounced when the agents within the team exhibit heterogeneity in their roles and capabilities. To develop a clearer understanding, let us consider the scenario of MMM2 (as shown in Fig. 1) within The StarCraft Multi-Agent Challenge (SMAC) environment [29]. In this setup, a team comprises 1 Medivac (responsible for healing teammates), 2 Marauders (focused on offensive actions), and 7 Marines (also dedicated to attacking). Each unit is controlled by a distinct agent, and the team's objective is to eliminate all enemy units. Learning an effective strategy for the Medivac unit proves particularly challenging due to its unique sub-task and limited number. Consequently, the individual experiences of the Medivac unit are deemed more likely to provide valuable insights for learning.

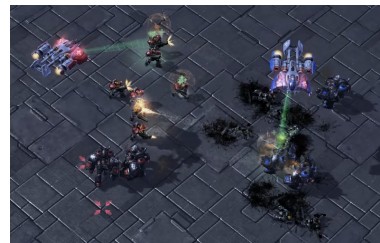

Figure 1: Screenshots of the MMM2 scenario in SMAC, highlighting the importance of individual experiences for the Medivac unit, given its unique sub-task and limited number.

Unfortunately, extracting these individual experiences from the team experiences proves difficult in the aforementioned MARL setting, as the individual rewards associated with each individual experience are unavailable.

In this paper, we introduce a novel method called **D**ecomposing **I**ndividual Reward **f**or **F**air **E**xperience **R**eplay (DIFFER), to address the above challenges associated with multi-agent reinforcement learning (MARL) experience replay. The fairness in MARL experience replay refers to the equitable sampling of individual experiences based on their importance. DIFFER addresses this by decomposing team experiences into individual experiences, facilitating a fair experience replay mechanism. We begin by proposing a method to calculate the individual reward for each individual experience, ensuring that DIFFER maintains equivalence with the original learning framework when the individual experiences are indistinguishable. To achieve this, we establish a partial differential equation based on the invariance of agent network gradients. By solving this differential equation, we obtain an approximation function for the individual rewards. Consequently, the team experience can be successfully decomposed into individual experiences. These individual experiences are then fed into a fair experience replay, where they are sampled for training the agent networks. The sampling probability is determined proportionally to the temporal difference (TD) error of each individual experience. Moreover, it adapts effectively to enhance both efficiency and fairness in scenarios where diversity is observed. Importantly, DIFFER is a generic method that can be readily applied to various existing value factorization techniques, thereby extending its applicability in MARL research.

We evaluate DIFFER on a diverse set of challenging multi-agent cooperative tasks, encompassing StarCraft II micromanagement tasks (SMAC) [29], Google Football Research (GRF) [30], and Multi-Agent Mujoco [31] tasks. Our experimental findings demonstrate that DIFFER substantially enhances learning efficiency and improves performance.

## 2   Preliminaries

**Problem Formulation.**   A fully cooperative multi-agent problem can be described as a decentralized partially observable Markov decision process (Dec-POMDP) [32], which consists of a tuple $G =<I, S, A, P, r, \Omega, O, N, \gamma >$. $I$ is the finite set of $N$ agents. For each time step, each agent $i \in I := \{1, \cdots, N\}$ receives individual observation $o_i$ according to the observation function $O(s, i) \equiv S \times I \rightarrow \Omega$. Then, each agent determines its action with individual policy $\pi_i(a_i|o_i) : \Omega \times A \rightarrow [0, 1]$. Given the team action $\boldsymbol{a} = \{a_1, \cdots, a_N\}$, the environment transits to a next state according to the

transition function $P(s'|s, \boldsymbol{a}) : S \times A^N \times S \to [0, 1]$ and gives a team reward $R(s, \boldsymbol{a}) \equiv S \times A^N \to \mathbb{R}$, which is shared by all agents. Each team policy $\boldsymbol{\pi}(\boldsymbol{a}|\boldsymbol{o}) = [\pi_1(a_1|o_1), ..., \pi_N(a_N|o_N)]$ has a team action-value function $Q_{\text{tot}}^t(\boldsymbol{o}^t, \boldsymbol{a}^t, s^t) = \mathbb{E}_{s_{t+1:\infty}, \boldsymbol{a}_{t+1:\infty}}[\text{Return}^t(s^t, \boldsymbol{a}^t)]$, where $\text{Return}^t(s^t, \boldsymbol{a}^t) = \sum_{k=0}^{\infty} \gamma^k R^{t+k}$ is a discount return and $\gamma$ is the discount factor. The target of a Dec-POMDP problem is to maximize the accumulated return of the team policy, *i.e.* optimal team policy $\boldsymbol{\pi}^* = \arg\max_{\boldsymbol{\pi}} \text{Return}(\boldsymbol{\pi}) = \mathbb{E}_{s_0 \frown d(s_0), \boldsymbol{a} \frown \boldsymbol{\pi}}[Q_{\text{tot}}^t(\boldsymbol{o}^t, \boldsymbol{a}^t, s^t)]$, where $d(s_0)$ represents the distribution of initial state.

**Multi-Agent Deep Q-learning.** Deep Q-learning [33] uses a neural network parameterized by $\theta$ to represent the action-value function $Q$. In the multi-agent Dec-POMDP problem, multi-agent deep Q-learning methods usually adopt the replay buffer [34] to store experience $(\boldsymbol{o}, \boldsymbol{a}, \boldsymbol{o}', R)$. Here $R$ is the team reward received after taking team action $\boldsymbol{a}$ with team observation $\boldsymbol{o}$, and $\boldsymbol{o}'$ represents team observation in the next step. During the training process, the parameter $\theta$ is updated by minimizing the Temporal Difference (TD) error with a mini-batch of data sampled from the replay buffer, which is shown as $L_{\text{TD}}(\theta) = \mathbb{E}_{(\boldsymbol{o}, \boldsymbol{a}, r, \boldsymbol{o}') \in D}[(r + \gamma \widetilde{Q}(\boldsymbol{o}'; \theta^-) - Q(\boldsymbol{o}, \boldsymbol{a}; \theta))^2]$, where $\widetilde{Q}(\boldsymbol{o}'; \theta^-) = \max_{\boldsymbol{a}'} Q(\boldsymbol{o}', \boldsymbol{a}'; \theta^-)$ is the one-step expected future return of the TD target and $\theta^-$ is the parameter of the target network [35]. $\delta = |r + \gamma \widetilde{Q}(\boldsymbol{o}'; \theta^-) - Q(\boldsymbol{o}, \boldsymbol{a}; \theta)|$ is known as TD-error.

**Multi-Agent Value Factorization.** The value factorization framework comprises $N$ agent networks (one for each agent) and a mixing network. At time step $t$, the agent network of agent $i$ takes the individual observation $o_i^t$, thereby computing the individual action-value, denoted as $Q_i$. Subsequently, the individual action-values $\{Q_i\}_{i \in I}$ are fed into the mixing network, which calculates the team action-value, denoted as $Q_{\text{tot}}$. The mixing network is constrained by Individual-Global-Max (IGM) principal[1], ensuring the congruity between the optimal team action deduced from $Q_{\text{tot}}$ and the optimal individual actions derived from $\{Q_i\}_{i \in I}$. The $N$ agent networks share parameters. We denote $\theta_m$ and $\theta_p$ as the parameters of mixing network and agent networks, respectively. During the training phase, the mixing network is updated based on the TD loss of $Q_{\text{tot}}$, while the agent networks are updated through gradient backward propagation. During the execution phase, each agent takes action with individual policy derived from its agent network in a decentralized manner.

# 3 Method

In this section, we present DIFFER, a novel experience replay method designed to foster fairness in multi-agent reinforcement learning (MARL) by decomposing individual rewards. DIFFER offers a solution to enhance the adaptivity of MARL algorithms. Instead of relying solely on team experiences, DIFFER trains the agent networks using individual experiences. This approach allows for a more granular and personalized learning process.

At a given time step, agents interact with the environment and acquire a **team experience** $\chi^{\text{team}} = (\boldsymbol{o}, \boldsymbol{a}, \boldsymbol{o}') = ((o_i)_{i \in I}, (a_i)_{i \in I}, (o_i')_{i \in I}, R) = ((o_i, a_i, o_i')_{i \in I}, R)$. Here, $o_i$, $a_i$, and $o_i'$ represent the observation, action, and next observation for each agent $i \in I$, respectively. These team experiences are stored in a replay buffer and subsequently sampled to train the mixing network and agent network parameters. We aim to decompose a multi-agent team experience into $N$ single-agent **individual experiences** $\{\chi_i^{\text{ind}}\}_{i \in I} = \{(o_i, a_i, o_i', r_i)\}_{i \in I}$, then utilize these individual experiences to update the agent networks.

The DIFFER framework comprises two stages: (1) **Decomposer**: Calculating individual rewards to decompose team experiences; (2) **Fair Experience Replay**: Selecting significant individual experiences to train the agent networks.

## 3.1 Exploring Individual Rewards through Invariance of Gradients

The DIFFER framework necessitates individual rewards $\{r_i\}_{i \in I}$ to decompose the team experience $\chi^{\text{team}}$ into individual experiences $\{\chi_i^{\text{ind}}\}_{i \in I}$. However, in the context of shared team reward POMDP problems, these individual rewards are not readily available. Therefore, the key challenge for DIFFER is to devise a strategy for calculating individual rewards.

In the traditional experience replay methods, agent networks are updated by the team TD-loss, which is calculated based on team experience and denoted as $L_{\text{tot}} = (R + \gamma \widetilde{Q}_{\text{tot}}(\boldsymbol{o}', \boldsymbol{a}'; \theta^-) - Q_{\text{tot}}(\boldsymbol{o}, \boldsymbol{a}; \theta))^2$.

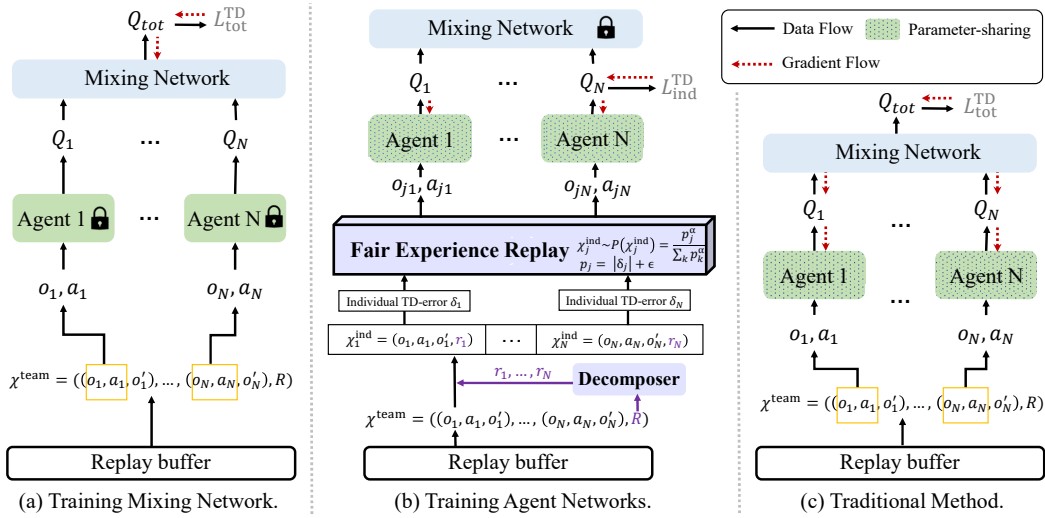

Figure 2: (a) In DIFFER, the mixing network is trained using the team TD-loss $L_{\text{tot}}^{\text{TD}}$, while the parameters of the agent networks remain unchanged. (b) The decomposing individual experiences are inputted into Fair Experience Replay, from which they are sampled to calculate the individual TD-loss $L_{\text{ind}}^{\text{TD}}$ for training the agent networks. (c) In traditional MARL experience replay methods, the mixing network and agent networks are updated using the team TD-loss $L_{\text{tot}}^{\text{TD}}$ derived from team experiences.

However, in DIFFER, agent networks are updated using the individual TD-loss, which is calculated based on individual experience and denoted as $L_i = (r_i + \gamma \widetilde{Q}_i(o_i', a_i'; \theta_p^-) - Q_i(o_i, a_i; \theta_p))^2$. To preserve the overall optimization objective of value factorization methods, it is desirable for an ideal individual reward approximation strategy to maintain invariance between the gradients of $L_{\text{tot}}$ (before decomposition) and the sum of $L_i$ (after decomposition) with respect to the parameters of the agent networks. The optimization equivalence can be formulated as follows:

$$\textbf{(Invariance of Gradient)} \quad \frac{\partial L_{\text{tot}}}{\partial \theta_p} = \sum_{i \in I} \frac{\partial L_i}{\partial \theta_p}, \tag{1}$$

where $\theta_p$ represents the parameters of the agent networks (noting that the agent networks of each agent share parameters). It is important to emphasize that this discussion focuses solely on the gradient of a single team experience. By solving the above partial differential equation, we are able to obtain an individual reward function that satisfies the invariance of gradients.

**Proposition 1.** *The invariance of gradient in Equ.(1) is satisfied when individual reward of agent $i$ is given by:*

$$r_i = (R + \gamma \widetilde{Q}_{\text{tot}} - Q_{\text{tot}}) \frac{\partial Q_{\text{tot}}}{\partial Q_i} - \gamma \widetilde{Q}_i + Q_i. \tag{2}$$

*for any agent $i \in I$.*

A rigorous proof and analysis of above proposition can be found in Appendix. Therefore, we approximate the individual rewards $r_i$ for individual experiences $\chi_i^{\text{ind}}$ while preserving the original optimization objective. In this way, a team experience $\chi^{\textbf{team}}$ is decomposed into $N$ individual experiences $\{\chi_i^{\text{ind}}\}_{i=1}^N$ successfully.

### 3.2 Fair Experience Replay Guided by Individual Experiences

After decomposition, we obtain a set of individual experience $E = \{\chi_j^{\text{ind}}\}_{j=0}^{N \cdot B}$ from a team experience mini-batch. Here, $j$ denotes the index of the individual experience in the set, $B$ represents the mini-batch size, and $N$ indicates the number of agents involved. Our objective is to construct a fair experience replay, a method that selects significant individual experiences from $E$ to train the agent network. Similar to Priority Experience Replay (PER) method [28], the TD-error of each individual experience is used as a metric to measure its significance. A larger TD-error for an individual

---

**Algorithm 1** Fair Experience Replay.

---

**Require:** The agent set $I$, the maximum steps $t_{\max}$, the mini-batch and target network update period $M$;
 1: Initialize replay memory $D$, set $t_{\text{step}} = 0$;
 2: Initialize the mixing network, agent network and target network with random parameters;
 3: **while** $t_{\text{step}} \leq t_{\max}$ **do**
 4:     For each episode, observe initial centralized observation $\{o_i^0\}_{i \in I}$;
 5:     set $t = 0$;
 6:     **while** Not terminal **do**
 7:         With probability $\epsilon$ select a random action $a_i^t$, otherwise $a_i = \arg\max_{a_i^t} Q_i(o_i^t, a_i^t)$ for each agent $i$;
 8:         Take action $\boldsymbol{a}^t$, then receive team reward $R_t$ and observation $\{o_i^{t+1}\}_{i \in I}$;
 9:         $t \leftarrow t + 1$;
10:     **end while**
11:     $t_{\text{step}} \leftarrow t_{\text{step}} + t$;
12:     Insert the current episode data into buffer $D$;
13:     Sample a random episode mini-batch of team experiences $\boldsymbol{B}^{\textbf{team}}$ from $D$;
14:     Calculate total TD-loss $L_{\text{TD}}^{\text{tot}}$ as Equ. (6) and update parameters of mixing network;
15:     Calculate individual reward as Equ. (8) and decompose team experience mini-batch $\boldsymbol{B}^{\textbf{team}}$ to individual experience set $E$ ;
16:     Calculate sample probability as Equ. (3) of each individual experience in mini-batch $E$;
17:     Calculate the sample ratio $\eta_{t_{\text{step}}}$ and sample significant individual experiences from $E$;
18:     Calculate individual TD-loss $L_{\text{TD}}^{\text{ind}}$ as Equ. (7) and update parameters of agent networks;
19:     Update target network parameters with period $M$.
20: **end while**

---

experience indicates that it is more valuable for the current model, and thus more likely to be selected for training. The individual experience $\chi_j^{\text{ind}}$ is sampled with probability:

$$P(\chi_j^{\text{ind}}) = \frac{p_j^{\alpha}}{\sum_k p_k^{\alpha}}, \tag{3}$$

where $p_j = |\delta_j| + \epsilon$. $\delta_j$ is the TD-error of $\chi_j^{\text{ind}}$ and $\epsilon$ is a small positive constant in case TD-error is zero. The hyper-parameter $\alpha$ determines the degree of prioritization. It degenerates to uniform sample cases if $\alpha = 0$. To correct the bias, $\chi_j^{\text{ind}}$ is weighted by importance-sampling (IS) weights as follows:

$$\omega_j = \left(\frac{1}{N} \cdot \frac{1}{P(\chi_j^{\text{ind}})}\right)^{\beta} / \max_k \omega_k. \tag{4}$$

The hyper-parameter $\beta$ anneals as introduced in PER.

In cases where the agents within a team are homogeneous, their individual experiences tend to exhibit similar TD-errors. As a result, these individual experiences are assigned similar probabilities during the fair experience replay process. In such circumstances, the fair experience replay approach employed by DIFFER effectively degenerates into a traditional method that utilizes team experiences.

Let $E'$ denote the selected individual experience set, which is a subset of the original individual experience set $E$. We define the sample ratio as $\eta := \#E'/\#E$, where $\#E'$ and $\#E$ are the numbers of individual experiences in $E'$ and $E$, respectively. Since the model is initially at a low training level and Q-value estimation errors are large, only a small portion of the individual experiences is worth training. However, this portion increases as the training progresses. Thus, motivated by the *warm-up* technique proposed in [36], we set $\eta < 1.0$ at the beginning of training and gradually increase it as the number of training steps increases. The sample ratio $\eta_{t_i}$ for a given training step $t_i$ can be expressed as follows:

$$\eta_{t_i} = \begin{cases} \eta_{\text{start}} + (\eta_{\text{end}} - \eta_{\text{start}})\dfrac{t_i}{pt_{\max}}, & t_i < pt_{\max} \\ \eta_{\text{end}}, & t_i \geq pt_{\max} \end{cases} \tag{5}$$

Here, the hyper-parameters $\eta_{\text{start}}$ and $\eta_{\text{end}}$ denote the initial and final values of the sample ratio, respectively. The hyper-parameter $p$ is the proportion of the time steps at which the sample ratio

increases, and $t_{\max}$ is the overall number of training steps. The ablation studies regarding the *warm-up* technique are provided in Appendix.

### 3.3 Overall Training and Evaluation

During the training phase, we first sample a team experience mini-batch $\boldsymbol{B^{\text{team}}}$ from the replay buffer. The parameters of mixing network $\theta_m$ are optimized with $L_{\text{tot}}$ calculated on the mini-batch by

$$L_{\text{TD}}^{\text{tot}}(\theta_m) = \sum\nolimits_{\chi^{\text{team}}=(o,a,o',R)\in B^{\text{team}}} (R + \gamma \cdot \widetilde{Q_{\text{tot}}}(o', \theta_m^-) - Q_{\text{tot}}(o, a; \theta_m))^2. \quad (6)$$

Next, we approximate the optimization equivalence of individual rewards of each individual experience as Equ. (8) and decompose a team experience into multiple individual experiences. Among them, we select a significant individual experience set $E'$ with probability in Equ. (3). The parameters of agent networks $\theta_p$ are optimized by

$$L_{\text{TD}}^{\text{ind}}(\theta_p) = \sum\nolimits_{\chi_j^{\text{ind}}=(o_j,a_j,o'_j,r_j)\in E'} \omega_j(r_j + \gamma \cdot \widetilde{Q_j}(o'_j; \theta_p^-) - Q_j(o_j, a_j; \theta_p))^2, \quad (7)$$

where $\omega_j$ is a weight assigned to individual experience $\chi_j^{\text{ind}}$ as calculated in Equ. (4).

During the inference phrase, agent $i$ chooses a greedy action according to its individual action-value $Q_i$ for decentralized execution. Therefore, the DIFFER framework meets centralized training and decentralized execution. The overall training and evaluation are presented in Algo. 1.

## 4 Experiments

In this section, we conduct several experiments to answer the following questions: (1) Are the decomposing individual rewards calculated using Equ. (8) optimization equivalent to the team reward experimentally? (Sec. 4.2) (2) Can DIFFER improve the performance of existing value factorization methods compared to the baselines across different environments? (Sec. 4.3) (3) Can DIFFER determine a more reasonable agent policy compared to the baselines? (Sec. 4.3) (4) Can DIFFER successfully distinguish and exploit the important individual experiences? (Sec. 4.4)

For every graph we plot, the solid line shows the mean value and the shaded areas correspond to the min-max of the result on 5 random seeds. All the experiments are conducted on a Ubuntu 20.04.5 server with Intel(R) Xeon(R) Gold 6248R CPU @ 3.00GHz and GeForce RTX 3090 GPU. The code is available in https://github.com/cathyhxh/DIFFER.

### 4.1 Experiments Setting

**Training Environments.** We choose discrete action space environments *The StarCraft Multi-Agent Challenge* (SMAC) [29] and *Google Football Research* (GRF) [30], as well as a continuous action space environment *Multi-Agent Mujoco* (MAMujoco)[31] to conduct our experiments. SMAC is currently a mainstream cooperative multi-agent environment with partial observability. We use the default environment setting for SMAC with version SC 2.4.10. GFR is a platform that facilitates multi-agent reinforcement learning in the field of football. We conduct the experiments on two academy scenarios in GRF. MAMujoco is a continuous cooperative multi-agent robotic control environment. Each agent represents a part of a robot or a single robot. All of the tasks mentioned in this work are configured according to their default configuration.

**Base Models.** We select QMIX [1], COMIX [31], QPLEX [2] and MASER [37] as our base models for comparison. QMIX and QPLEX are renowned value factorization methods that have been widely used in discrete action space environments. COMIX is an extension of QMIX specifically designed for continuous action space environments. These three base models employ team experience to train their overall value factorization frameworks, as illustrated in Fig. 2(c). Furthermore, we include MASER, a MARL experience replay method, in our evaluation. Similar to our methods, MASER generates individual rewards and trains agent networks using individual experiences.

**Implementation Hyper-parameters.** All base models were implemented by faithfully adhering to their respective open-source codes based on PyMARL. Regarding the $warm\_up$ trick for the sample

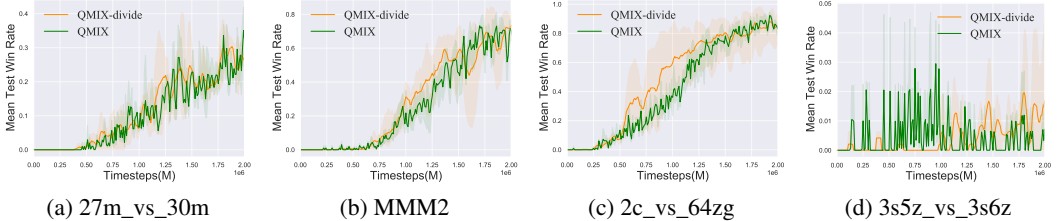

(a) 27m_vs_30m      (b) MMM2      (c) 2c_vs_64zg      (d) 3s5z_vs_3s6z

Figure 3: Performance comparison of QMIX-divide and QMIX on SMAC scenarios, highlighting the optimization equivalence between team reward and individual reward calculated by DIFFER.

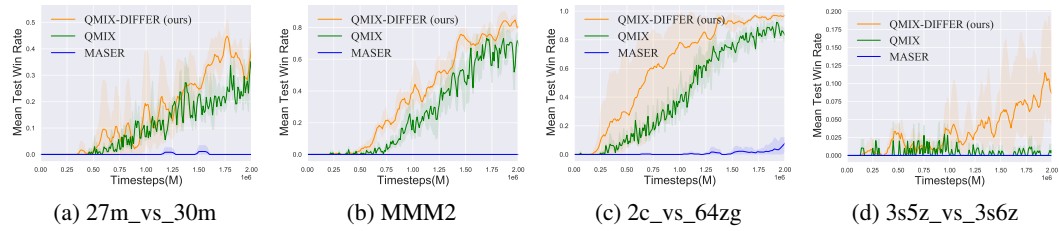

(a) 27m_vs_30m      (b) MMM2      (c) 2c_vs_64zg      (d) 3s5z_vs_3s6z

Figure 4: Performance comparison of QMIX-DIFFER (ours) and QMIX on SMAC scenarios, highlighting the performance improvement of our method towards QMIX.

ratio, we initially set the ratio to 0.8, which gradually increases linearly until reaching 1.0 at 60% of the total training steps. From that point onwards, the sample ratio remains fixed at 1.0 for the remainder of the training process. For a comprehensive overview of the hyper-parameter settings employed in our experiments, we refer readers to the appendix.

## 4.2 Optimization Equivalence of the Individual Experiences

In this section, our objective is to investigate whether the decomposition of individual experiences in DIFFER yields an equivalence to team experiences within the original learning framework. To accomplish this, we focus on the QMIX as the base model, which updates agent networks using the team TD-loss computed from team experiences. We introduce a variant model named **QMIX-divide**. In QMIX-divide, the agent networks are updated using the individual TD-loss calculated from **all** decomposed individual experiences. Similar to QMIX, the mixing network of QMIX-divide is updated using the team TD-loss. Compared with DIFFER, QMIX-divide preserves the decomposer of team experiences but omits this selection phase in the fair experience replay. Our aim is to test the hypothesis that the decomposed individual experiences have no impact on agent network optimization. We compare the performance of QMIX-divide and QMIX on SMAC scenarios, as depicted in Fig. 3. Our results demonstrate that QMIX-divide and QMIX exhibit nearly identical performance, providing evidence in support of the optimization equivalence between the team reward and the approximated individual reward. Furthermore, we highlight the critical role played by the selection phase in the fair experience replay, as the omission of this phase in QMIX-divide leads to comparable performance with the baselines QMIX.

## 4.3 Performance Comparison

**Performance Improvement towards Value Factorization Methods.** We investigated whether DIFFER could enhance the performance of established value factorization methods. To evaluate this, we conducted experiments using QMIX, QPLEX, and COMIX as the base models in various environments. The training curves for each model are depicted in Fig. 4, Fig. 5, Fig. 6, and Fig. 7. As illustrated in the figures, DIFFER exhibited notable improvements in both learning speed and overall performance compared to the baselines during the training process. The magnitude of improvement varied across different scenarios. Notably, DIFFER demonstrated a more significant performance boost in scenarios where agents possessed distinct physical structures (*e.g.*, agents with different

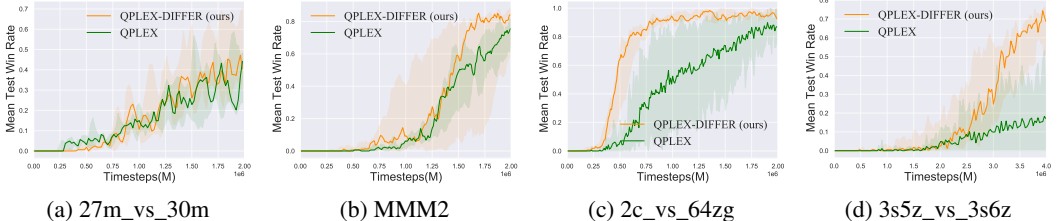

(a) 27m_vs_30m  (b) MMM2  (c) 2c_vs_64zg  (d) 3s5z_vs_3s6z

Figure 5: Performance comparison of QPLEX-DIFFER (ours) and QPLEX on SMAC scenarios, highlighting the performance improvement of our method towards QPLEX.

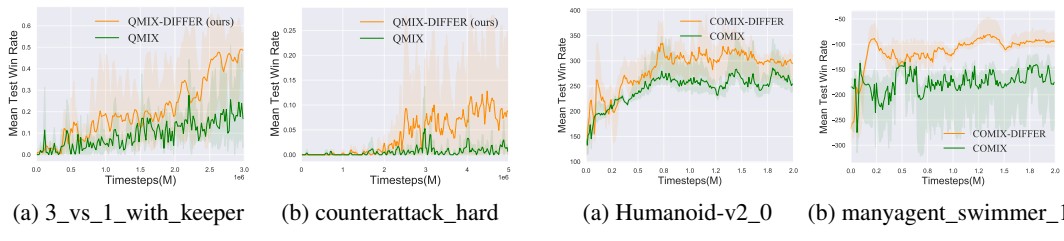

(a) 3_vs_1_with_keeper  (b) counterattack_hard

Figure 6: Performance comparison of QMIX-DIFFER (ours) and QMIX on GRF scenarios, highlighting the performance improvement of our method towards QMIX.

(a) Humanoid-v2_0  (b) manyagent_swimmer_1

Figure 7: Performance comparison of COMIX-DIFFER (ours) and COMIX on MAMujoco scenarios, highlighting the performance improvement of our method towards COMIX.

joints in Humanoid − v2_0) or subtasks (*e.g.*, agents with different skills in 3s5z_vs_3s6z). This observation aligns with our hypothesis that scenarios with greater dissimilarity among agents result in a more pronounced improvement through the distinction of individual experiences facilitated by DIFFER. Conversely, in scenarios where agents were homogeneous (*e.g.*, 27m_vs_30m in SMAC), DIFFER exhibited performance similar to that of the base model. In such cases, the lack of significant differences among agents rendered the decomposition of individual experiences less impactful. Overall, our findings demonstrate that DIFFER effectively enhances the performance of established value factorization methods. The degree of improvement depends on the specific characteristics of the scenarios, emphasizing the importance of considering the heterogeneity of agents when applying DIFFER.

**Comparison with MARL ER methods.** In Fig. 4, we present a comparison of the performance of DIFFER and MASER. While both models implement individual rewards and leverage individual experiences to update agent networks, they differ in their optimization objectives. Specifically, DIFFER retains the original objective of maximizing the team action-value $Q_{tot}$. In contrast, MASER aims to maximize a mixture of individual action-value $Q_i$ and team action-value $Q_{tot}$, which alters the original optimization objective. As a consequence of this shift in optimization objective, the performance of MASER is observed to be worse than DIFFER in challenging environments.

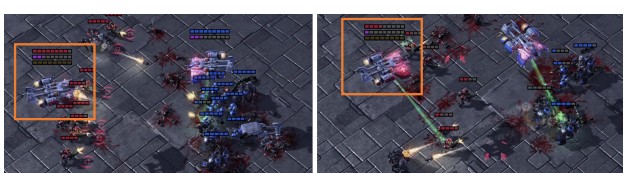

Figure 8: Screenshots of QMIX (left) and QMIX-DIFFER (right) on SMAC scenario MMM2. The red team is controlled by a trained model, while the blue team is controlled by a built-in AI. The Medivac in the red team is marked by an orange square. The red team is controlled by QMIX in the left subfigure and by QMIX-DIFFER in the right subfigure.

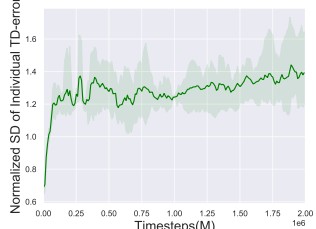

Figure 9: The normalized standard deviation (std) of TD-error of individual experiences.

**Case Study.** The screenshot in Fig. 8 shows the game screen for the QMIX and QMIX-DIFFER models in the SMAC scenario MMM2. As previously noted, learning an effective strategy for the medivac unit is particularly challenging due to its distinct sub-task and a limited number of units. A successful medivac strategy must incorporate a viable movement strategy that avoids enemy attacks and reduces unnecessary damage, as well as a sound selection of healing objects that maximizes team damage output. In the QMIX controlled agent team (Fig. 8, left), the medivac failed to learn an appropriate movement strategy, moving too close to the enemy team and dying early in the episode. As a result, the other agents bled out quickly and the episode ended in defeat. In contrast, the agent team controlled by QMIX-DIFFER (Fig. 8, right) demonstrated an effective medivac strategy, positioning the unit in a safe location to avoid enemy attacks and providing healing support throughout the game. The resulting advantage for the red team led to a final victory.

### 4.4 Analysis for TD-error of Individual Experiences

In this section, we delve into the differentiation of individual experiences by analyzing their TD-errors. Throughout the training process, we compute the standard deviation (std) of the TD-errors of the decomposing individual experiences and normalize them using the mean TD-error. Figure 9 showcases the training curve of the normalized std of individual TD-errors. Notably, we observe a substantial increase in the normalized std at the initial stages of training, which remains consistently high until the conclusion of the training process. This observation underscores the substantial distinctions that exist among individual experiences when evaluated based on their TD-errors. Our DIFFER framework, by decomposing individual rewards, effectively captures and exploits these distinctions among individual experiences. Consequently, DIFFER exhibits superior performance compared to the baseline models. These findings highlight the efficacy of DIFFER in leveraging the differentiation among individual experiences to enhance learning outcomes in MARL settings.

### 4.5 Visualization of Individual Rewards Produced by DIFFER

In order to provide a clear visual representation of the individual rewards acquired through our proposed DIFFER method, we introduce game screenshots alongside the corresponding individual rewards for each agent, as illustrated in Fig. 10. For the purpose of analysis, we consider two distinct timesteps of an episode within the MMM2 scenario. Our team comprises 2 Marauders (index 0-1), 7 Marines (index 2-8), and a Medivac (index 9). The team rewards assigned to agents are proportional to the damage inflicted on enemy units and the number of enemy unit deaths. During the initial stage of the episode (timestep 4, (a)), our units possess sufficient health and do not require assistance from the Medivac. Consequently, the contribution of the Medivac to the team's performance is minimal. As a result, the individual reward attributed to the Medivac is relatively lower compared to the Marauders and Marines, who play active roles in offensive opera-

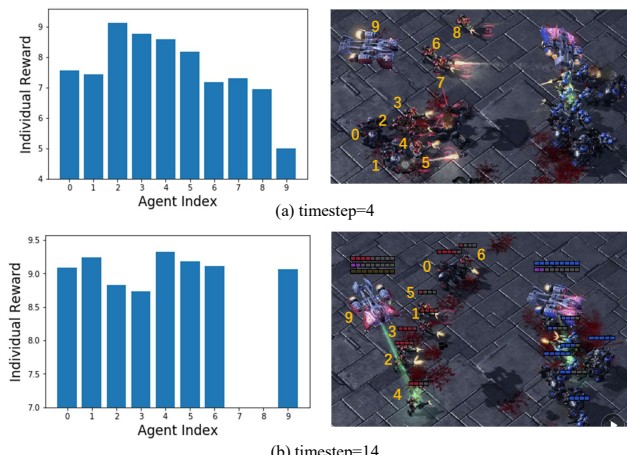

(a) timestep=4

(b) timestep=14

Figure 10: The visualization of individual rewards produced by our DIFFER method, along with game screenshots captured at timestep 4 and timestep 14 of the MMM2 scenario. The index of each agent have been marked in orange number. The consistent correlation between individual rewards and the ongoing game situation serves as a testament to the efficacy and rationality of our DIFFER method.

tions. Agents 2-5, being in closer proximity to the enemy and inflicting substantial damage, enjoy higher individual rewards due to their effective engagement strategies. At timestep 14 (b), except for agent 7 and 8 who have perished, the individual rewards assigned to each agent are substantially balanced. This indicates a comparable level of contribution from all agents towards achieving team objectives at this particular point in the game. The persistent correlation between individual rewards

and the dynamic in-game circumstances not only reaffirms the effectiveness of our DIFFER method but also validates its rationality.

# 5 Related Work

Experiment Replay (ER) is a widely used mechanism for improving data utilization and accelerating learning efficiency in reinforcement learning (RL). ER reuses the historical experience data for updating current policy [34, 38]. In multi-agent RL, using a single-agent ER algorithm directly to obtain individual experience is naive if agents can obtain accurate individual rewards from the environment [39, 40]. However, in the multi-agent problem setting addressed in this work, a team of agents only has access to a shared team reward, making it impossible to obtain an accurate individual reward. Consequently, multi-agent ER algorithms employ joint transitions as minimal training data [41, 1, 2]. MASER[37] is a method similar to ours, as it calculates individual rewards from the experience replay buffer and uses individual experience to train the individual value network. However, MASER generates individual rewards to maximize the weighted sum of individual action-value and team action-value, which may violate the original optimization target of multi-agent RL, *i.e.*, maximizing the team action-value. In our work, we propose a novel strategy for approximating individual reward and demonstrate its optimization equivalence. To our knowledge, DIFFER is the first work to approximate individual reward in a way that ensures optimization equivalence.

# 6 Conclusion

**Limitation.** A limitation of the DIFFER method is the introduction of additional computation when calculating the individual reward and TD-error. This increased computational cost is an inherent trade-off for the benefits gained in terms of fairness and improved learning efficiency. The additional computational burden should be considered when applying this method in resource-constrained settings or scenarios with strict real-time requirements.

In conclusion, this paper presents the DIFFER framework. By decomposing the team reward into individual rewards, DIFFER enables agents to effectively distinguish and leverage important individual experiences. Through the invariance of network gradients, we derive a partial differential equation that facilitated the computation of the underlying individual rewards function. By incorporating the solved closed-form individual rewards, we calculate the individual temporal-difference error, providing the significance of each experience. Notably, DIFFER showcases its adaptability in handling both homogeneous and diverse individual experiences, demonstrating its versatility in various scenarios. Our extensive experiments on popular benchmarks demonstrate the effectiveness of DIFFER in terms of learning efficiency and fairness, surpassing the performance of existing MARL ER methods. We hope that our work provides a new perspective on experience replay methods in MARL, inspiring further research and advancements in the field.

# Acknowledgments

This work is supported by National Key R&D Program of China under Contract 2022ZD0119802, and National Natural Science Foundation of China under Contract 61836011. It was also supported by GPU cluster built by MCC Lab of Information Science and Technology Institution, USTC, and the Supercomputing Center of the USTC.

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

# A The Proof of Proposition 1

**Proposition 1.** *The invariance of gradient is satisfied when individual reward of agent $i$ is given by:*

$$r_i = (R + \gamma \widetilde{Q}_{\text{tot}} - Q_{\text{tot}}) \frac{\partial Q_{\text{tot}}}{\partial Q_i} - \gamma \widetilde{Q}_i + Q_i. \tag{8}$$

*for any agent $i \in I$.*

*Proof.* According to the chain rule,

$$\frac{\partial L_{\text{tot}}}{\partial \theta_p} = \frac{\partial L_{\text{tot}}}{\partial Q_{\text{tot}}} \sum_{i \in I} \frac{\partial Q_{\text{tot}}}{\partial Q_i} \frac{\partial Q_i}{\partial \theta_p} = 2(R + \gamma \widetilde{Q}_{\text{tot}} - Q_{\text{tot}}) \sum_{i \in I} \frac{\partial Q_{\text{tot}}}{\partial Q_i} \frac{\partial Q_i}{\partial \theta_p}; \tag{9}$$

$$\sum_{i \in I} \frac{\partial L_i}{\partial \theta_p} = \sum_{i \in I} \frac{\partial L_i}{\partial Q_i} \frac{\partial Q_i}{\partial \theta_p} = \sum_{i \in I} 2(r_i + \gamma \widetilde{Q}_i - Q_i) \frac{\partial Q_i}{\partial \theta_p}. \tag{10}$$

Therefore,

$$\frac{\partial L_{\text{tot}}}{\partial \theta_p} = \sum_{i \in I} \frac{\partial L_i}{\partial \theta_p},$$

$$\Longleftrightarrow 2(R + \gamma \widetilde{Q}_{\text{tot}} - Q_{\text{tot}}) \sum_{i \in I} \frac{\partial Q_{\text{tot}}}{\partial Q_i} \frac{\partial Q_i}{\partial \theta_p} = \sum_{i \in I} 2(r_i + \gamma \widetilde{Q}_i - Q_i) \frac{\partial Q_i}{\partial \theta_p},$$

$$\Longleftrightarrow 2 \sum_{i \in I} [(R + \gamma \widetilde{Q}_{\text{tot}} - Q_{\text{tot}}) \frac{\partial Q_{\text{tot}}}{\partial Q_i}] \frac{\partial Q_i}{\partial \theta_p} = 2 \sum_{i \in I} (r_i + \gamma \widetilde{Q}_i - Q_i) \frac{\partial Q_i}{\partial \theta_p}, \tag{11}$$

$$\Longleftarrow (R + \gamma \widetilde{Q}_{\text{tot}} - Q_{\text{tot}}) \frac{\partial Q_{\text{tot}}}{\partial Q_i} = r_i + \gamma \widetilde{Q}_i - Q_i, \forall i \in I$$

$$\Longleftrightarrow r_i = (R + \gamma \widetilde{Q}_{\text{tot}} - Q_{\text{tot}}) \frac{\partial Q_{\text{tot}}}{\partial Q_i} - \gamma \widetilde{Q}_i + Q_i, \forall i \in I.$$

In conclusion, a sufficient condition for invariance of gradients is as follow: individual reward of agent $i$ is calculated as Equ. (8) for any $i \in I$. □

# B The Hyper-parameters Setting

In DIFFER, the replay buffer stores the most recent 5000 episodes, and mini-batches of size 32 are sampled from it. The target network is updated every 200 episodes. RMSProp is utilized as the optimizer for both the mixing network and agent network, with a learning rate of 0.0005, $\alpha$ (decay rate) set to 0.99, and $\epsilon$ (small constant) set to 0.00001. Gradients are clipped within the range of [-10, 10]. The discount factor for the expected reward (return) is 0.99.

Regarding the fair experience replay, we apply the "warm-up" technique to the sample ratio $\eta$. It linearly increases from 0.8 to 1.0 over 60% of the time steps and remains constant thereafter. The regulation parameter $\alpha$, which determines the prioritization degree in fair experience replay, is set to 0.8. The regulation parameter $\beta$ of the sampling probability anneals linearly from 0.6 to 1.0.

For exploration, we employ an $\epsilon$-greedy strategy with $\epsilon$ linearly annealed from 1.0 to 0.05 over 50K time steps and then held constant for the remainder of the training. The maximum time step during training is set to 2005000 for all experiments. Each scenario is run with 5 different random seeds.

The agent network architecture follows a DRQN[27] structure, consisting of a GRU layer for the recurrent layer with a 64-dimensional hidden layer. Before and after the GRU layer, there are fully-connected layers with 64 dimensions each. The mixing network is implemented using open-source code. All experiments on the SMAC benchmark adopt the default reward and observation settings provided by the benchmark. For the baseline algorithms, we use the authors' code with hyper-parameters fine-tuned specifically for the SMAC benchmark.

The supplementary materials contain the code for our DIFFER framework. The main codes of individual reward calculation and fair experience replay are shown in

DIFFER_code/learners/q_learner_divide.py and DIFFER_code/ER/PER/prioritized_memory.py, respectively.

The specific hyper-parameters are listed below.

Table 1: The hyper-parameters that used in our DIFFER framework.

| Hyper parameters | Meaning | Value |
|---|---|---|
| epsilon_start | Start value of $\epsilon$ anneal | 1.0 |
| epsilon_end | End value of $\epsilon$ anneal | 0.05 |
| epsilon_anneal_time | Duration step of $\epsilon$ anneal | 50000 |
| mempool_size | Memory pool size in learner | 5000 |
| batch_size | Batch size per training | 32 |
| target_update_interval | Interval steps between 2 updates of target network | 200 |
| lr | Learning rate | 0.0005 |
| regularization | Weights of L1-regularizer | 0.0005 |
| tot_optimizer | Optimizer in training | RMSProp |
| tot_optim_alpha | Alpha of optimizer | 0.99 |
| tot_optim_eps | Epsilon of optimizer | 0.00001 |
| ind_optimizer | Optimizer in training | RMSProp |
| ind_optim_alpha | Alpha of optimizer | 0.99 |
| ind_optim_eps | Epsilon of optimizer | 0.00001 |
| grad_norm_clip | Clip range of gradient normalization | 10 |
| $\eta_{start}$ | Start value of $\eta$ warm up | 0.8 |
| $\eta_{end}$ | End value of $\eta$ warm up | 1.0 |
| $p$ | Duration step ratio of $\eta$ warm up | 0.6 |
| $\alpha$ | Regulation parameter of sampling probability | 0.8 |
| $\epsilon$ | Regulation parameter of sampling probability | 0.01 |
| $\beta_{start}$ | Start value of $\beta$ warm up | 0.6 |
| $\beta_{end}$ | End value of $\beta$ warm up | 1.0 |

## C  Ablation Studies for warm_up Technique of Sample Ratio

To highlight the significance of the $warm\_up$ trick for the individual experience sample ratio $\eta$, we conduct ablation experiments on SMAC maps `MMM2`. Fig. 11 showcases the performance of QMIX-DIFFER with and without the $warm\_up$ sample ratio. In this context, the model **W-warm_up (ours)** refers to the setting where the sample ratio linearly increases as the training progresses. On the other hand, **W/O-warm_up** denotes the scenario where the sample ratio remains fixed throughout the training phase. To ensure a fair comparison, we conduct experiments with different fixed values of $\eta$, while keeping the batch-size of all three models in Fig. 11 consistent at 32. Let us assume that the overall quantity of training data is $K$ for W/O-warm_up-$\eta$=1.0. By employing the hyper-parameter setting detailed in Sec. B, we can calculate that the overall training data amounts to $0.94K$ for W-warm_up. Consequently, we select $\eta$ from the set $\{0.8, 0.94, 1.0\}$. The results clearly demonstrate that the $warm\_up$ trick leads to performance improvements, even when using a smaller amount of training data. This highlights the importance of incorporating the $warm\_up$ strategy into the training process.

## D  Analysis of Sampling Percentage

In this section, we present the distribution of individual experience sampling percentages across different agent classes within the `MMM2` scenario from the SMAC environment[29]. The team composition consists of three distinct unit classes: 1 Medivac unit responsible for healing teammates, 2 Marauder units specialized in offensive actions, and 7 Marine units dedicated to attacking. Throughout the training process, DIFFER selects important individual experiences from a diverse set of 10 agents to update the agent networks. We calculate the sampling percentages of individual experiences for

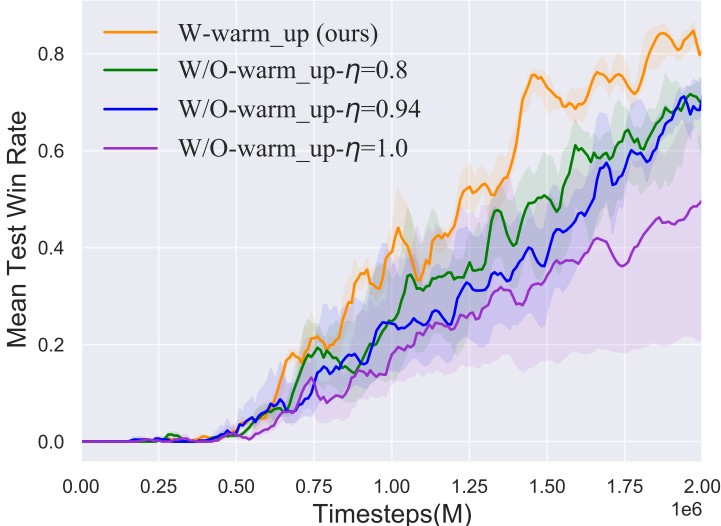

Figure 11: Ablation studies regarding the *warm up* trick of individual transition sample ratio on `MMM2`. "W-warm_up (ours)" represents the sample ratio increases as the training step increases until 1.0. "W/O-warm_up" represents the sample ratio $\eta$ is fixed during training.

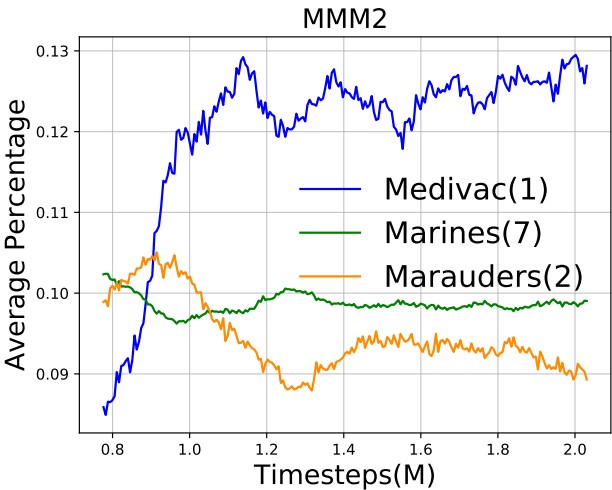

Figure 12: Average percentage of sampled individual experiences from each unit class on SMAC scenario `MMM2`. The number in the label represents the quantity of unit. The sampling percentage of a unit class is high when it poses a learning challenge.

each agent and derive the average sampling percentage for each unit class. The corresponding curves, focusing on the later stages of training for improved stability, are depicted in Fig. 12.

The graph showcases significant variations in the average sampling percentages among the unit classes, reflecting their distinct learning difficulties. As the learning difficulty of a particular unit class intensifies, a larger proportion of individual experiences from that class are sampled during training. This adaptive behavior of the DIFFER method highlights its ability to dynamically adjust the sampling percentages to address the unique learning challenges encountered by each unit class. Notably, achieving effective strategies for the Medivac unit proves particularly challenging due to its distinctive sub-task and limited availability. Consequently, the average sampling percentage of the Medivac unit surpasses that of the other two unit classes significantly. Although the number of Marine units exceeds that of Marauder units, the Marines are encouraged to adopt a specialized positioning strategy. This strategy requires the Marines to spread out in a fan-like formation, aiming

to minimize the damage inflicted upon them while maximizing their effectiveness in dealing damage to the enemy. Consequently, the Marines face higher learning difficulties compared to the Marauders. Reflecting this distinction, the average sampling percentage of the Marine unit class surpasses that of the Marauder unit class.

# E Discussion Related to Policy-Based Methods

## E.1 Performance Comparison with Policy-Based Method COMA

In this section, we compare DIFFER with the classic policy-based method COMA[3] on four SMAC scenarios as shown in Fig. 13. We can see that DIFFER performs better than COMA on all four scenarios, which strongly highlight the strengths and advantages of our proposed DIFFER framework.

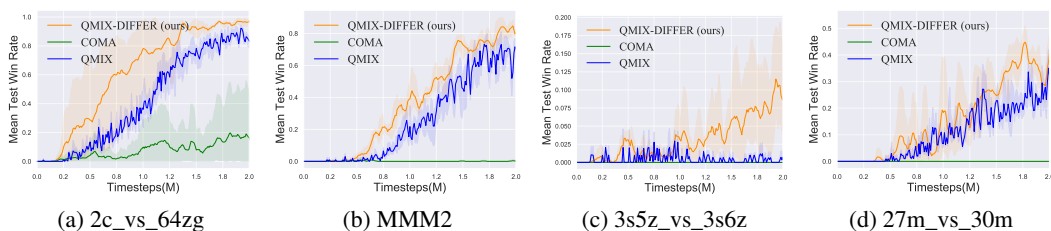

| (a) 2c_vs_64zg | (b) MMM2 | (c) 3s5z_vs_3s6z | (d) 27m_vs_30m |

Figure 13: Performance comparison of QMIX-DIFFER (ours) and a classic policy-based method COMA on SMAC scenarios, highlighting the performance improvement of our method towards COMA.

## E.2 Potential Application to Policy-Based Approaches

Our method, DIFFER, is specifically designed to address MARL problems where individual rewards are not available, and teams only have access to a shared team reward. In such scenarios, calculating individual action-values becomes a challenge. However, DIFFER provides a solution by decomposing individual trajectories, enabling the calculation and update of individual action-values. For policy-based methods that approximate individual action-values using individual critics, such as IPPO[42] and MADDPG[43], the traditional approach involves treating the team reward as an individual reward and using it to update individual critics. In this context, the decomposition of team experiences provided by DIFFER may not be directly applicable or necessary since the individual critics can approximate action-values using the team reward. On the other hand, for policy-based methods that do not require the approximation of individual action-values, such as MAPPO[4], our DIFFER method might not offer significant enhancements. In such cases, where there is no need to decompose team trajectories for individual action-values training, the contribution of DIFFER may not be as pronounced.

# F Environment Introduction

In this section, we provide a brief overview of the multi-agent environment discussed in this paper.

## F.1 SMAC

The Starcraft Multi-Agent Challenge (SMAC)[29] (as shown in Fig. 14 (a)) is a prominent cooperative multi-agent environment known for its partial observability. In this paper, we utilize the default environment setting of SMAC, specifically version SC2.4.10. SMAC offers a variety of scenarios, each featuring a confrontation between two teams of units. One team is controlled by a trained model, while the other team is governed by a built-in strategy. The scenarios differ in terms of the initial unit positions, the number and types of units in each team, as well as the characteristics of the map terrain. In the SMAC environment, a team is deemed victorious when all units belonging to the

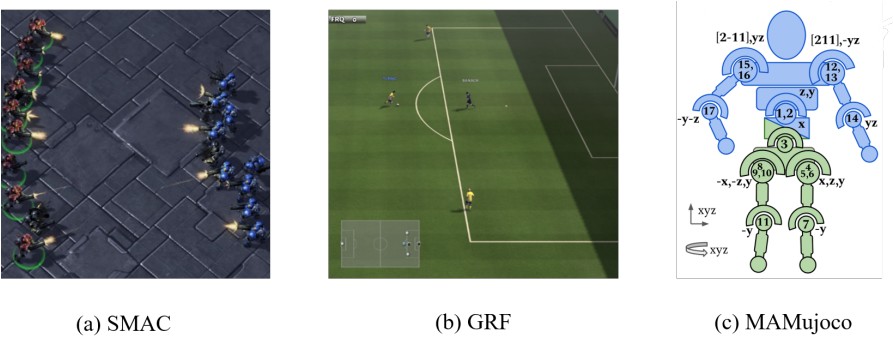

| (a) SMAC | (b) GRF | (c) MAMujoco |
|---|---|---|

Figure 14: Visualisation of three experimental environments.

opposing team are eliminated. The primary objective of the policy is to maximize the win rate across all scenarios. To facilitate training, the environment provides a shaped reward function that takes into account factors such as hit-point damage inflicted and received by agents, the number of units killed, and the outcome of the battle. Learning a coordinated micromanagement strategy in diverse maps poses a significant challenge for the agents in this environment. Such a strategy aims to maximize the damage inflicted upon enemies while minimizing the damage sustained. Overall, the SMAC environment demands agents to acquire sophisticated coordination and decision-making skills in order to excel in diverse scenarios and effectively apply micromanagement strategies.

## F.2 GRF

Google Research Football (GRF) environment [30] (shown in Fig. 14 (b)) presents a challenging multi-agent reinforcement learning (MARL) setting where a team of agents must learn to pass the ball amongst themselves and overcome their opponents' defense to score goals. Similar to the SMAC environment, the opposing team in GRF is controlled by a built-in strategy. In GRF, each agent has 19 different actions at their disposal, including standard move actions in eight directions, as well as various ball-kicking techniques such as short and long passes, shooting, and high passes that are difficult for opponents to intercept. The agents' observations encompass information about their own position, movement direction, the positions of other agents, and the ball. An episode in GRF terminates either after a fixed number of steps or when one of the teams successfully scores a goal. The winning team receives a reward of +100, while the opposing team is rewarded with -1. In this study, we specifically focus on two official scenarios: `academy_3_vs_1_with_keeper` and `academy_counterattack_hard`.

## F.3 MAMujoco

The Multi-Agent Mujoco (MAMujoco) environment[31] (shown in Fig. 14 (c)) is a highly versatile and realistic simulation platform designed for multi-agent robotic control tasks. It is based on the Mujoco physics engine and offers continuous action spaces, allowing for smooth and precise control of robotic agents. Each agent in the environment represents a specific component or a complete robot, enabling the simulation of complex multi-robot systems. All of the tasks mentioned in this work are configured according to their default configuration. We set maximum observation distance to $k = 0$ for $Humanoid - v2$ and $k = 1$ for `manyagent_swimmer`.

We also provide the description of agent types in each scenario in Table 2. It aims to enhance the reader's understanding of the distinct performance improvements brought about by DIFFER.

Table 2: The agent type distribution in each scenario.

| Env. | Scenario Name | #Agent | #Type | Type Distribution |
|---|---|---|---|---|
| SMAC | 27m_vs_30m | 27 | 1 | 27 |
| | 2c_vs_64zg | 2 | 1 | 2 |
| | MMM2 | 10 | 3 | 1-2-7 |
| | 3s5z_vs_3s6z | 8 | 2 | 3-5 |
| GRF | academy_3_vs_1_with_keeper | 3 | 2 | 1-2 |
| | academy_counterattack_hard | 4 | 2 | 2-2 |
| MAMujoco | Humanoid-v2_0 | 2 | 2 | 1-1 |
| | manyagent_swimmer | 20 | 10 | 2-2-2-2-2-2-2-2-2-2 |

