# OpenReview forum: "DIFFER:Decomposing Individual Reward for Fair Experience Replay in Multi-Agent Reinforcement Learning"
_NeurIPS.cc/2023/Conference — NeurIPS 2023 poster_

### Official Review · Reviewer_ddQi · 2023-06-30

**Soundness:** 3 good
**Presentation:** 2 fair
**Contribution:** 3 good
**Rating:** 5
**Confidence:** 5

**Summary:**

This paper proposes to train a mixing network and agent networks separately for multiagent reinforcement learning when the system is based on the centralized training and decentralized execution (CTDE) paradigm. The authors derive an individual reward that maintains invariance between the gradients of the mixing network's loss function and the sum of the loss functions of the agent networks. Then, the authors apply Prioritized Experience Replay to select samples for training the agent networks. The proposed method, DIFFER, is compared with QMIX, COMIX, QPLEX, and MASER on several tasks, such as SMAC, GRF, and MAMujoco. The experimental results show that DIFFER outperforms the baseline methods and demonstrate that a fair experience replay mechanism works efficiently.

**Strengths:**

- Originality: The idea of fair experience replay based on the constraint between the gradients of the mixing and the agent networks is novel. However, I found the following paper that may be related to this study: D. Hostallero et al. (2020). Inducing Cooperation through Reward Reshaping based on Peer Evaluations in Deep Multi-Agent Reinforcement Learning. In Proc. of AAMAS.
- Quality: The experimental results support the claims and the proposed method.
- Clarity: The manuscript is written very well and easy to follow.
- Significance: While the CTDE approach becomes popular in MARL, many previous studies do not pay attention to the difference between individual experiences. This study sheds light on its importance.


**Weaknesses:**

- Although the experimental results show that the proposed method works well, there is no theoretical analysis. In particular, the gradient of the loss function of the agent network is independent of the target networks as described later. It would be better to discuss the loss function in detail.
- In addition, the off-policyness of the proposed method should be discussed.


**Questions:**

Major comments:
- The individual reward (2) is interesting, but I am unsure how the TD error is calculated. I think that the TD error is given by
  $$
  (R + \gamma \tilde{Q}_{\mathrm{tot}} - Q_{\mathrm{tot}}) \frac{\partial Q_{\mathrm{tot}}}{\partial Q_i}
  $$
  because the term $\gamma \tilde{Q}_i – Q_i$ cancels out. Is the target network $\tilde{Q}_i$ necessary? Is my understanding correct?
- I would like to know whether the used learning algorithm is off-policy or not. Experience replay buffer technique is typically applied to off-policy reinforcement learning algorithms, but the loss functions (6) and (7) suggest that the action value function is trained in an on-policy manner, like SARSA. Is on-policyness necessary for the proposed method?
- Is the input of the mixing network consistent with the outputs of the agent networks when the mixing and agent networks are trained separately?
- Does QMIX-divide adopt uniform sampling instead PER for training the agent networks? If so, the performance difference between QMIX-DIFFER and QMIX-divide can be discussed based on Fujimoto et al. (2020).
S. Fujimoto et al. (2020). An Equivalence between Loss Functions and Non-Uniform Sampling in Experience Replay. NeurIPS 33.

Minor comments:
- Line 87: $A^n$ -> $A^N$.
- Line 96: $r$ -> $R$.
- Line 154: Since $S$ denotes the state space, it would be better to use a different symbol.
- Eq.(7) and Line 191: $\chi^{\mathrm{indi}}_j$ -> $\chi^{\mathrm{ind}}_j$.



**Limitations:**

Minor comments:
The authors do not discuss this work's potential negative social impacts.

---

> ### Author Rebuttal · Authors · 2023-08-09
>
> Dear Reviewer:
>
> Thanks for your valuable comments.
>
> **Q1: Is $\widetilde{Q_i}$ necessary?**
>
> A1: Your understanding is indeed correct.
>
> 1) In the computation of the TD error for the agent networks, there is no need to explicitly determine the value of $\widetilde{Q_i}$ because it cancels out within the formula.
> The TD error can be expressed as $(R + \gamma \widetilde{Q} _ {\mathrm{tot}}-Q_{\mathrm{tot}})\frac{\partial Q_{\mathrm{tot}}}{\partial Q_i}$, as you have correctly stated.
>
> 2) However, it is worth noting that the calculation of $\widetilde{Q_i}$ remains necessary for the determination of $\widetilde{Q_{\mathrm{tot}}}$. The target mixing network receives the $\widetilde{Q_i}$ values from each individual agent and produces the $\widetilde{Q_{\mathrm{tot}}}$ output.
>
> 3) While it may not be necessary to explicitly calculate individual rewards using Equation (2) in the code, incorporating this calculation can significantly enhance our understanding and improve interpretability.
>
> **Q2: Is the used learning algorithm off-policy?**
>
> A2: Yes.
>
> 1) We would like to clarify that the action value functions in our work are trained in an off-policy manner similar to Q-learning , contrary to an on-policy approach like SARSA.
> This is evident in Line 100, where we define $\widetilde{Q}(\boldsymbol{o’};\theta^-)$ as the optimal target action value, represented as $\max_{\boldsymbol{a’}}Q(\boldsymbol{o’},\boldsymbol{a’};\theta^-)$.
>
> 2) As a result, in Equation 6, $\widetilde{Q_{\mathrm{tot}}}(\boldsymbol{o’},\theta_m^-)=\max_{\boldsymbol{a’}} Q_{\mathrm{tot}}(\boldsymbol{o’},\boldsymbol{a’};\theta_m^-)$ represents the optimal target team action value function, demonstrating the off-policy nature of our learning algorithm.
>
> 3) Regarding Equation 7, we apologize for the typo.
> The part of $\widetilde{Q_{j}}(o'_j, a_j;\theta_p^-)$ is incorrect,
>
> which should be $\widetilde{Q_{j}}(o'_j;\theta_p^-)$.
>
> The correct expression of Equation 7 should be
>
> $$L^{\mathrm{ind}} _ {\mathrm{TD}}(\theta_p)=\sum\nolimits_{\chi^{\mathrm{indi}} _ j=(o_j,a_j,o_j',r_j)\in S'}\omega_j(r_j +\gamma \cdot \widetilde{Q_{j}}(o'_j;\theta_p^-)-Q_j(o_j,a_j;\theta_p))^2$$
>
> We regret any confusion stemming from our oversight and appreciate the opportunity to rectify it in the revised version of our paper.
>
> **Q3: Is the input of the mixing network consistent with the outputs of the agent networks?**
>
> A3: Yes. In Algorithm 1 on page 5, specifically in Line 13-19, we provide a detailed description of a network update procedure.
>
> 1) After retrieving data from the replay buffer, the mini-batch data is fed into the agent network, resulting in the computation of individual action-values ($Q_i$). These individual action-values are subsequently forwarded to the mixing network, which combines them to obtain the team action-value ($Q_{\mathrm{tot}}$).
>
> 2) We calculate both the individual TD-loss and the team TD-loss independently. Notably, the gradient propagation and network updating steps are carried out independently for both the agent networks and the mixing network.
>
> 3) In the revised version of our paper, we intend to emphasize the consistent flow of information between the agent and mixing networks throughout the training process.
>
> **Q4: The reason of the performance difference between QMIX-DIFFER and QMIX-divide.**
>
> A4:
> 1) QMIX-divide adopts all samples in the mini-batch instead of PER for training the agent networks.
>
> 2) We have thoroughly studied the paper you referred to and have identified that it indeed provides an explanation for our observed performance improvement to a certain degree. Consequently, we have decided to cite this paper in the revised version of our work and carefully discuss the implications it holds for our research.
>
> **Q5: Minor comments.**
>
> A5: We apologize for the presence of these typos and sincerely appreciate your valuable insights. We will promptly rectify the errors to ensure clarity and precision in our paper.
>
> Please let us know if you have any further concerns, and we are encouraged to have a discussion.

---

> > ### Author Response · Authors · 2023-08-17
> > **Be Glad to Tell Us Any Concern**
> >
> > Dear Reviewer:
> >
> > We genuinely value your suggestions and comments.
> > We address your comments point by point and try our best to respond to them.
> > If there are further suggestions or questions, please let us know them.
> > We are eager to engage in a discussion with you.
> >
> > Best regards,
> >
> > The authors of DIFFER

---

### Official Review · Reviewer_U3kJ · 2023-07-02

**Soundness:** 4 excellent
**Presentation:** 4 excellent
**Contribution:** 4 excellent
**Rating:** 7
**Confidence:** 3

**Summary:**

 This is a very interesting and strong paper on MARL that seeks the strategy of learning individual policies rather than team policies in order to break symmetry and learn identity and contextual influences on the individual policies. In order to do so, it solves the technical problem of  decomposing a team reward into individual reward, and provides a proof that their method works. They then employ a replay buffer that selects individual experiences to train the agent network akin to prioritized replay. The authors demosntrate their method on a variety of benchmarks, showing improvement.

**Strengths:**

The method is novel; the theory and experiments solid. The writing was very clear and flowed very naturally.

The authors demonstrate their method not just multi agent Mujoco, but even the difficult StarCraft II micromanagement tasks.

**Weaknesses:**

I do not have any concerns.

**Questions:**

N/A

**Limitations:**

The authors have a limitations section that is fair and does not weaken the strengths of the paper.

---

> ### Author Rebuttal · Authors · 2023-08-09
>
> Dear Reviewer:
>
> Thanks for your valuable comments. We sincerely appreciate your positive evaluation and recognition of our research.
>
> To enhance our work’s robustness and comprehensiveness, we plan to incorporate additional experiment results, intuitive visualization, and theoretical analysis in the revised edition of our paper. These enhancements aim to strengthen the validity, clarity, and impact of our findings. Your valuable feedback has been instrumental in shaping the development of our research, and we are grateful for your contribution.

---

### Official Review · Reviewer_f7XM · 2023-07-06

**Soundness:** 3 good
**Presentation:** 3 good
**Contribution:** 3 good
**Rating:** 6
**Confidence:** 4

**Summary:**

This paper introduces a method for decomposing shared rewards into individual rewards within value-based multi-agent reinforcement learning (MARL) methods. By decomposing the rewards, the method enables the prioritization of crucial individual experiences. Experimental results demonstrate the effectiveness of the proposed approach across a range of value-based MARL methods.

**Strengths:**

1. The paper provides theoretical evidence for the optimization equivalence between value-based MARL methods and those employing decomposed rewards.

2. The concept of decomposing shared rewards into individual rewards is innovative and appears to yield positive results, showcasing the potential for significant impact.

3. The authors have made their code available, ensuring reproducibility of the experiments.

**Weaknesses:**

1. The paper lacks a discussion of policy-based MARL methods, which is a significant gap in the related work. It would be valuable to include a comprehensive overview and comparison with policy-based approaches.

2. The absence of a comparison with policy-based methods diminishes the completeness of the evaluation. Including such a comparison would further strengthen the paper's contribution. Some related papers are [1][2].

**Questions:**

1. Is there any intuitive visualization or illustration available that can help understand the decomposed individual rewards more clearly?

2. How does this method relate to policy-based MARL methods? It would be beneficial to discuss the potential application of individual reward decomposition to policy-based approaches.

**Limitations:**

The paper primarily focuses on cooperative MARL tasks. It would be intriguing to explore the applicability of reward decomposition in mixed cooperative-competitive settings.

---

> ### Author Rebuttal · Authors · 2023-08-09
>
> Dear Reviewer:
>
> Thank you for the constructive comments.
>
> **Q1: Intuitive visualization of the decomposed individual rewards.**
>
> A1: We have included visualizations displaying the individual rewards for each agent, along with corresponding screenshots, which can be found in the Figure 3 of Rebuttal PDF. The presented visualizations demonstrate that the individual rewards yielded by DIFFER align with the current game situation, further supporting the effectiveness of our approach. To provide greater clarity on the concept and implications of decomposed individual rewards, we will incorporate it in the revised version of our paper.
>
> **Q2: How does this method relate to policy-based MARL methods?**
>
> A2:
> 1) *Performance Comparison with Policy-Based Methods:*
>
>      We notice that you may dismiss the related paper details in Weaknesses 2, so that we don’t know which exactly the related papers you mentioned. We have conducted a comprehensive experiment comparing DIFFER with the classic policy-based method COMA[1]. The detailed results of this comparison can be found in the “global” response PDF, which strongly highlight the strengths and advantages of our proposed DIFFER framework.
>
> 2) *Potential Application to Policy-Based Approaches:*
>
>      Our method, DIFFER, is specifically designed to address MARL problems where individual rewards are not available, and teams only have access to a shared team reward. In such scenarios, calculating individual action-values becomes a challenge. However, DIFFER provides a solution by decomposing individual trajectories, enabling the calculation and update of individual action-values.
>
>       a) For policy-based methods that approximate individual action-values using individual critics, such as IPPO[2] and MADDPG[3], the traditional approach involves treating the team reward as an individual reward and using it to update individual critics. In this context, the decomposition of team experiences provided by DIFFER may not be directly applicable or necessary since the individual critics can approximate action-values using the team reward.
>
>       b) On the other hand, for policy-based methods that do not require the approximation of individual action-values, such as MAPPO[4], our DIFFER method might not offer significant enhancements. In such cases, where there is no need to decompose team trajectories for individual action-values training, the contribution of DIFFER may not be as pronounced.
>
> In our forthcoming research, we will embark on an exploration of the application of our innovative method, DIFFER, within the domain of policy-based methods.
>
> Please share any remaining concerns.
>
> [1] Foerster, Jakob, et al. "Counterfactual multi-agent policy gradients." Proceedings of the AAAI conference on artificial intelligence. Vol. 32. No. 1. 2018.
>
> [2] de Witt, Christian Schroeder, et al. "Is independent learning all you need in the starcraft multi-agent challenge?." arXiv preprint arXiv:2011.09533 (2020).
>
> [3] Yu, Chao, et al. "The surprising effectiveness of ppo in cooperative multi-agent games." Advances in Neural Information Processing Systems 35 (2022): 24611-24624.
>
> [4] Lowe, Ryan, et al. "Multi-agent actor-critic for mixed cooperative-competitive environments." Advances in neural information processing systems 30 (2017).

---

> > ### Author Response · Authors · 2023-08-17
> > **Be Glad to Tell Us Any Concern**
> >
> > Dear Reviewer:
> >
> > We appreciate your comments and time!
> > We wonder whether our response have response all of your questions about our work.
> > And we are most glad to discuss anything with you about our work.
> >
> > Best regards,
> >
> > The authors of DIFFER

---

> > > ### Comment · Reviewer_f7XM · 2023-08-18
> > >
> > > Thanks for your responses. I would maintain my recommendation.

---

### Official Review · Reviewer_Jgve · 2023-07-21

**Soundness:** 3 good
**Presentation:** 3 good
**Contribution:** 3 good
**Rating:** 6
**Confidence:** 3

**Summary:**

The paper presents a new algorithm DIFFER to help individual agents to learn individual TD-error such that it improves the overall (team) model performance. The paper's proposed method is different from an existing method MASER in that it is maintaining the overall team objective. Overall model performance comparison was made against QMIX which uses the traditional team TD-error only.

**Strengths:**

The paper adequately points out the limitation of the proposed method in that additional complexity thus more resources is required. The paper also demonstrates how each complexity (i.e. individual TD-error computation and the selection phase) contributes to the final model performance compared to a prior art. This helps the reader make an informed decision based on the tradeoff between complexity and utility demonstrated. The paper seems overall well-written such that it was fairly easy to follow.

**Weaknesses:**

I would like to see more comparisons of DIFFER's performance and complexity against MASER's, a closely related work.

**Questions:**

See the weakness section above.

**Limitations:**

The authors adequately listed the potential limitation in the paper.

---

> ### Author Rebuttal · Authors · 2023-08-09
>
> Dear Reviewer:
>
> Thanks for your valuable suggestion.
>
> 1) We conducted experiments to compare the performance of DIFFER and MASER in SMAC environments. According to your suggestion, we have conducted additional experiments to meticulously evaluate the performance of DIFFER in relation to MASER on the Google Football Research (GRF) task, which can be found in the Figure 1 of Rebuttal PDF. The comprehensive results underscore the strengths and advantages of our proposed DIFFER framework.
>
> 2) Here we explain the factors leading to the poor performance of MASER.
>
>     a) It is important to acknowledge that MASER introduces a divergence in the optimization objective by seeking to maximize a combination of individual action-value and team action-value, as explained in lines 264-270. This divergence may contribute to its limitations.
>
>     b) It is crucial to note that MASER predominantly focuses on addressing sparse reward multi-agent reinforcement learning (MARL) scenarios, whereas our experiments are exclusively conducted within dense reward environments.
>
> Please let us know if you have any further concerns, and we are encouraged to have a discussion.

---

> > ### Author Response · Authors · 2023-08-17
> > **Be Glad to Tell Us Any Concern**
> >
> > Dear Reviewer:
> >
> > We sincerely appreciate your valuable feedback.
> > We wonder whether our response adequately addresses all of your concern.
> > We are more than willing to provide any additional clarification or answer any questions you may have regarding our work.
> >
> > Best regards,
> >
> > The authors of DIFFER

---

### Author Rebuttal · Authors · 2023-08-09

We extend our sincere gratitude to all the esteemed reviewers for their invaluable and constructive comments.
The consistently positive reviews our submission received from all the Reviewers have filled us with delight.
We are greatly encouraged by their recognition of the originality (Review f7XM, U3kJ, ddQi), significant impact (Review U3kJ, ddQi), theoretical foundation (Review f7XM, U3kJ) , reproducibility (Review f7XM), and clarity (Review Jgve, U3kJ, ddQi) showcased in our work. Their meticulous evaluation of our research has further bolstered our confidence in the merits and contributions of our study.

We also appreciate reviewers pointing out our weaknesses. We address their comments point by point and try our best to respond to them. Hope our response addresses the reviewers' concerns.

The additional experiments in the Rebuttal PDF are summarised as follows:

In Rebuttal Figure 1, we show a comparison between  our model DIFFER and MASER[1] on Google Football Research (GRF) tasks, highlighting the performance improvement of our method towards MASER..

In Rebuttal Figure 2, we show a comparison between our model DIFFER and a classic policy-based method COMA[2], highlighting the performance improvement of our method towards COMA..

In Rebuttal Figure 3, we present the visualization of individual rewards produced by our DIFFER method. The consistent correlation between individual rewards and the ongoing game situation serves as a testament to the efficacy and rationality of our DIFFER method.

[1] Jeon, Jeewon, et al. "Maser: Multi-agent reinforcement learning with subgoals generated from experience replay buffer." International Conference on Machine Learning. PMLR, 2022.

[2] Foerster, Jakob, et al. "Counterfactual multi-agent policy gradients." Proceedings of the AAAI conference on artificial intelligence. Vol. 32. No. 1. 2018.

---

### Decision · Program_Chairs · 2023-09-21

**Decision:**

Accept (poster)

**Comment:**

This paper has been positively evaluated by all reviewers. The rebuttal helped to clarify concerns about the theoretical guarantees of the method, the lack of discussion of related works, and the effectiveness of the proposed method. The scores did not change substantially after the rebuttal phase, and the general positive consensus remained.

I encourage the authors to address all the comments and to incorporate the recommended improvements in the final version.